# Advancement of NF-κB Signaling Pathway: A Novel Target in Pancreatic Cancer

**DOI:** 10.3390/ijms19123890

**Published:** 2018-12-05

**Authors:** Kartick C. Pramanik, Monish Ram Makena, Kuntal Bhowmick, Manoj K. Pandey

**Affiliations:** 1Department of Basic Sciences, Kentucky College of Osteopathic Medicine, University of Pikeville, Pikeville, KY 41501, USA; 2Department of Physiology, The Johns Hopkins University, School of Medicine, Baltimore, MD 21205, USA; mmakena1@jhmi.edu; 3Department of Biomedical Sciences, Cooper Medical School of Rowan University, Camden, NJ 08103, USA; bhowmickk8@rowan.edu (K.B.); pandey@rowan.edu (M.K.P.)

**Keywords:** NF-κB, PDAC, inflammation, resistance, prevention

## Abstract

Pancreatic ductal adenocarcinoma (PDAC) is one of the deadliest cancers and is the third highest among cancer related deaths. Despite modest success with therapy such as gemcitabine, pancreatic cancer incidence remains virtually unchanged in the past 25 years. Among the several driver mutations for PDAC, *Kras* mutation contributes a central role for its development, progression and therapeutic resistance. In addition, inflammation is implicated in the development of most human cancer, including pancreatic cancer. Nuclear factor kappa-light-chain-enhancer of activated B cells (NF-κB) is recognized as a key mediator of inflammation and has been frequently observed to be upregulated in PDAC. Several lines of evidence suggest that NF-κB pathways play a crucial role in PDAC development, progression and resistance. In this review, we focused on emphasizing the recent advancements in the involvement of NF-κB in PADC’s progression and resistance. We also highlighted the interaction of NF-κB with other signaling pathways. Lastly, we also aim to discuss how NF-κB could be an excellent target for PDAC prevention or therapy. This review could provide insight into the development of novel therapeutic strategies by considering NF-κB as a target to prevent or treat PDAC.

## 1. Introduction

The pancreas is a vital endocrine and exocrine organ that lies in the upper abdomen behind the stomach. The exocrine pancreas, which comprises more than 95% of the pancreatic mass, secretes digestive enzymes into the duodenum. It includes acinar and ductal cells associated with connective tissue, vessels, and nerves. The pancreatic juice produced by the exocrine pancreas flows into the common bile duct via the pancreatic duct, and empties into the duodenum to aid in digestion. The endocrine pancreas primarily consists of α and β cells which secrete glucagon, insulin, somatostatin and pancreatic polypeptide into the blood [1]. Considering the location and function of the pancreas, pancreatic cancer can have devastating effects. For example, pancreatic ductal adenocarcinoma (PDAC) arises exocrine cells. Almost 85% cases of pancreatic cancers are PDAC. The other forms of pancreatic cancers have been reported, nonetheless these are either less common or rare, such as neuroendocrine tumors, acinar carcinomas, pancreatoblastomas, colloid carcinomas, and solid-pseudopapillary neoplasms [1].

PDAC accounts for approximately 7.3% of all cancer deaths. Sadly, despite a steady increase in medium survival times for most cancers over the past few decades, survival rates of pancreatic cancer remain a formidable challenge, with the five-year survival of patients remaining as low as 8.5% in the USA [2]. According to recent cancer statistics, 55,440 new cases of pancreatic cancers will be detected and 44,330 people will die in 2018 in the USA [2]. Alarmingly, new cases of pancreatic cancer have been rising on an average of 0.5% annually over the past ten years, and it is expected to become the second leading cause of cancer-related death in the US by 2030 [3,4]. Unfortunately, the increase in mortality is not just restricted to USA; it is increasing worldwide [5].

Various factors contribute to low survival rates of PDAC. Amongst these are late stage diagnosis and an increasing prevalence in the elderly population. Almost 75% of the patients diagnosed are between 55–84 years of age and the median onset age is 71 years [6]. Additionally, metastasis also contributes in poor survival; almost 90% of patients are diagnosed with metastasis [7]. There is a genetic component and strong family history, with 10% to 15% of the PDAC cases exhibiting mutations in *BRCA2, BRCA1, CDKN2A, ATM, STK11, PRSS1, MLH1* and *PALB2* [7]. Furthermore, cigarette smoking is one of the major risk factors in the development of PDAC [8]. Additionally, diabetes, chronic pancreatitis, and obesity confer increased risk for PDAC [7,9,10].

Large-scale cancer genomic studies have shown a heterogeneous mutational profile in PDAC [11]. Almost 63 genetic alternations have been identified, with the most common *Ras* mutation. Almost 90% of PDAC are associated with *Ras* mutation. Inactivating mutations of *TP53, CDKN2A* and *SMAD4* showed up in 50–80% of pancreatic cancer cases. Other genes, including *ARID1A, MLL3* and *TGFBR2*, are mutated in ~10% of tumors [1,12].

There is a dearth of effective systemic therapies that also contributes to PDAC cancer mortality [7]. Surgery remains the only chance for cure of PDAC; however, most patients relapse and die. Also, at early diagnosis, few patients are found eligible for resection [13].

In 1997, gemcitabine (a nucleoside analogue), was approved to treat pancreatic cancer by the US FDA. Later in 2005, FDA approved a tyrosine kinase inhibitor of EGFR, erlotinib along with gemcitabine for pancreatic cancer treatment. Recently, phase 3 ACCORD-11 trial, using FOLFIRINOX regimen (oxaliplatin 85 mg/m², folinic acid [leucovorin] 400 mg/m², irinotecan 180 mg/m², bolus fluorouracil 400 mg/m², infusional fluorouracil 2400 mg/m² over 46 h, every 14 days) showed a better response, progression-free survival, and overall survival compared to gemcitabine monotherapy in patients with metastatic PDAC. Either FOLFIRINOX or the combination of combination of gemcitabine along with nanoparticle albumin–bound paclitaxel (nab-paclitaxel) is considered standard treatment for patients with PDAC. However, due to the side effects of this cocktail of chemotherapeutic drugs, only 30–40% of PDAC patients can be treated with the combination. Nonetheless, the side effect of chemotherapy remains the main obstacle in achieving quality of life of the patients [1,7,13]. As a result, there is an urgent need to extend our understanding about PDAC, in terms of pathways regulating its oncogenic role, and to develop novel therapeutic strategies in order to improve long term survival of PDAC patients.

## 2. NF-κB Signaling Pathways

The transcription factor, nuclear factor kappa-light-chain-enhancer of activated B cells (NF-κB) was first discovered in 1986 as a regulator of B-cell specific transcription factor. Since its discovery, it has been found to be involved in key processes and regulates the expression of number of genes involved in immune system, cell survival, stress responses, embryogenesis, differentiation, proliferation and cell death [14]. Because of its critical role in a variety of biological events, the dysregulation of NF-κB pathways can lead to various ailments including cancer, autoimmune diseases, neurodegenerative diseases, cardiovascular disease, and diabetes [14,15].

The NF-κB family consists of five master transcription factors, RelA/p65, RelB, c-Rel, and precursor proteins p50/p105 (NF-κB1) and p100/p52 (NF-κB2). The functional form NF-κB1 (p50) and NF-κB2 (p52) are produced by proteasome mediated processing of precursor proteins p105, and p100 respectively [16,17]. These transcription factors form various dimeric complexes and bind to κB sites, which induce or repress transcription. Out of various dimers, the well-studied dimer is p50/65. The activation of NF-κB is mediated by two pathways: classical/canonical, and non-classical/non-canonical. Any external stimulus including oxidative stress, viruses, bacteria, growth factors, lead to the activation of classical pathways where NF-κB is rapid and transiently activated. The classical pathways mainly involve p50–p65 dimers and play critical role in the regulation of gene products involved in angiogenesis, survival, metastasis, and cell proliferation. The first step in the classical activation of signaling is the activation of TGFβ activated kinase 1 (TAK1, or called as MAP3K7), followed by activation of trimeric IκB kinase (IKK) complex, which is made up of catalytic (IKKα and IKKβ) and regulatory (IKKγ, also called as NEMO) subunits. The IKK complex then phosphorylates inhibitory protein IκBα. The phosphorylation of IκBα leads to its ubiquitination and proteasomal degradation. The degradation of IκBα releases the dimer p50/65 in cytoplasm followed by nuclear translocation. Translocated complex binds with the specific sequence of DNA and regulates the gene expression [18,19]. Studies based on genetic mouse models suggest that IKKγ and IKKβ are critical in phosphorylation-dependent IκBα degradation, whereas IKKα plays a supporting role [20]. 

The alternative/non-canonical pathway depends on de novo synthesis of NF-κB-inducing kinase (NIK, also known as MAP3K14), which is activated by a sub group of cytokines such as lymphotoxin (LT), receptor activator of NF-κB ligand (RANKL/TNFSF11), CD40 ligand (CD40L), and B cell activating factor of the TNF family (BAFF/TNFSF13B). Based on genetic studies it has been established that NIK is required for non- canonical NF-κB activation [21]. NIK activates IKKα, which, mediates the processing of p100 [22]. The alternative pathway results in the activation of p52–RELB dimers and regulates the expression of genes associated with the maintenance of secondary lymphoid organs and tumor microenvironment [15,23,24].

## 3. Role of NF-κB Pathways in Cancer Development and Progression

Several agents including oxidative stress, pro-inflammatory cytokines, growth factors, viral and bacterial agents activate NF-κB (Figure 1). Once activated, NF-κB regulates key gene products playing critical roles in tumorigenesis. Aberrant NF-κB signaling has shown to be associated with a variety of tumors, including breast, colon, leukemia, lymphoma, lung, prostate, pancreatic, thyroid, and ovarian carcinoma [15,25].

### 3.1. Inflammation and Immune Modulation

The activation of canonical NF-κB signaling has been associated in inflammatory processes [26] (Figure 1). Tumor necrosis factor alpha (TNF-α) and IL-6 are two of the best-studied pro-inflammatory and pro-tumorigenic cytokines, the expression of which are elevated in many different cancers. Activated macrophages and neutrophils mainly produce TNF-α, IL-1 and IL-6 which may accelerate tumorigenesis. TNF signaling entails activation of NF-κB and mitogen-activated protein kinases (MAPKs), by activating transcription factor AP1, and promote cell survival of malignant cells. Additionally, IL-6 links NF-κB to STAT3, where this axis plays a prominent role in cancer-related inflammation and other pathologies [15,27]. 

Another key modulator of inflammation is reactive oxygen species (ROS), which are produced by mitochondrial and other cellular byproducts. The role of ROS has been reported in inflammatory disorders and in various stages of cancer development. Importantly, ROS can activate NF-κB by various mechanisms and paradoxically, NF-κB also regulates ROS. It has been demonstrated that NF-κB regulated genes play an important role in the detoxification of ROS. However, some gene products such as nitric oxide synthases (NOS) do perform a pro-oxidant function. ROS and reactive nitrogen species can also induce DNA damage and oncogenic mutations leading to cancer [25,28].

NF-κB signaling pathways is also critical in T and B cell development and proliferation. Thus, constitutive NF-κB activation leads to the continuous proliferation of lymphocytes which is the main culprit in leukemia and lymphoma [29]. NK cells exert anti-tumorigenic activity in leukemia and lymphomas by killing the malignant cells. NF-κB modulates the anti-tumorigenic potential of NK cells, by upregulating the expression of perforins and granzyme B, which inhibit lytic potentials of NK cells [15].

### 3.2. Proliferation of Cancer

NF-κB plays a major role in tumor initiation, tumor promotion and tumor progression (Figure 1). NF-κB activation promotes mutagenesis, at various stages such as in DNA strands, daughter cells, and mismatch repair genes, which eventually leads to genomic instabilities. Chronic inflammation and NF-κB can also induce chromosomal instability and aneuploidy leading to tumor initiation. NF-κB promotes cancer cell proliferation and survival, by enhancing the expression of various cyclins including cyclin D1 and cyclin D2 [15,24]. *Kras* and p53 mutations have been found in 20–25%, and in ~50% of all cancers, respectively. Oncogenic mutations in EGFR and PI3K also contribute to NF-κB activation and tumor proliferation [25].

### 3.3. Cell Death

One of the “hallmarks of cancer” is avoidance of cell death. Constitutive activation of NF-κB has been reported in variety of cancer types including pancreatic cancers, which is associated with the drug resistance [30]. NF-κB was shown to regulate pro-and anti-apoptotic genes *cIAPs; TRAIL, caspase-8 and c-FLIP, BCL-2* and *BCL-XL* [25]. Tumor suppressor p53 activates pro-apoptotic proteins Puma and NOXA, leading to apoptosis. NF-κB and p53 antagonistically regulate each other’s activity, while IKKβ was shown to directly phosphorylate p53 and promote its polyubiquitylation and degradation [15]. In addition to p53 mediated apoptosis, studies have demonstrated that GTPase HRAS159 and c-MYC induced NF-κB activation inhibits cell death [31,32]. NF-κB and autophagy intricately regulate each other’s activity. For example, NF-κB family member p65/RelA was shown to upregulate protein expressions which are associated in autophagy such as beclin 1 and SQSTM1, whereas autophagy was shown to degrade IKK subunits and inhibit NF-κB signaling [33,34,35]. NF-κB induced chemo-resistance will be discussed in a later part of the review in the context of pancreatic cancer.

### 3.4. Tumor Microenvironment

Evolution of the tumor microenvironment occurs in three steps: niche construction, expansion, and maturation. It has been demonstrated that NF-κB is critical in all steps (Figure 1) [15]. Moreover, studies have further emphasized that activation non-canonical NF-κB signaling is crucial in tumor microenvironment [36]. Additionally, it has been demonstrated that cross interaction of STAT3 and NF-κB signaling pathways regulates the production of cytokines, growth factors which play an important role in the constitution of tumor microenvironment [27]. Furthermore, inhibition of NF-κB signaling in cancer associated fibroblasts (CAF’s) abolished its tumor-promoting effects, suggesting that NF-κB is critical in CAFs mediated tumor enhancing effects [15]. Additionally, NF-κB plays a key role in the constitution of tumor microenvironment in multiple myeloma. It is activated by diverse bone marrow-derived cytokines and growth factors, and also by direct physical contact between multiple myeloma cells and stromal cells [37].

### 3.5. Angiogenesis

The formation of new blood vessels from existing endothelial lined vessels, defined as angiogenesis, is crucial for tumor growth. The growth factors such as vascular endothelial growth factor (VEGF), fibroblast growth factor (FGF), and platelet-derived growth factor (PDGF) regulates angiogenesis. It has been demonstrated that NF-κB regulates angiogenesis. For example, inhibition of NF-κB activity inhibits the tumor growth and angiogenesis, in different cancer models, including pancreatic cancer [38,39,40,41,42]. Along these lines, κB sites have been in identified in promoter region of matrix metalloproteinases (MMPs) genes, which are involved in angiogenesis [24,43].

### 3.6. Metastasis

Approximately 90% of cancer deaths happen because of metastasis, which is the mobilization of cancer cells from primary tumor sites to either surrounding sites or distant organs. Epithelial–mesenchymal transition (EMT) is considered to be a hallmark of metastasis [44]. Blocking NF-κB inhibited the transcription of the EMT genes *CDH2, SLUG, TWIST1* and *SNAIL* in different cancer cell lines (Figure 1) [19,45,46,47,48]. Furthermore, it has been demonstrated that TNF-α induced NF-κB regulates expression of *TWIST1* in both normal breast epithelial and breast cancer cells leading metastatic phenotype [49]. In addition to regulating EMT genes, NF-κB can promote cell migration and invasion through other mechanisms. For example, NF-κB activation can stimulate the expression of hypoxia-inducible factor 1α (HIF1α), thereby may enhance hypoxic conditions and survival of metastasis-initiating cells [50].

### 3.7. Cancer Stem Cells

Cancer stem cells (CSCs) are immortal cells within a tumor that can self-renew and give rise to many cell types that constitute the tumor. CSCs contribute to metastasis and drug resistance in tumor cells. NF-κB signaling is an important pathway that regulates survival, activation and differentiation of CSCs (Figure 1). In fact, both classical and alternative activation of NF-κB may promote proliferation and survival of stem cells in various cancers [51]. In pancreatic CSCs, the classical NF-κB pathway appears to increase the expression of a transcription factor SOX9, which is critical in invasion [52].

## 4. NF-κB and Pancreatic Cancer

Almost 90% of pancreatic cancers are driven by oncogenic mutations of *Kras.* Aberrant *Kras* signaling activates inflammatory signaling pathways that play critical roles in regulating the initiation of pancreatic intraepithelial neoplasia (PanIN) and the progression of PDAC [53]. The fact that *Kras* is involved in the pancreatic cancer development is established by studies of several genetic mouse models. Along these lines, several mouse models have been developed showing the direct link of IKK and *Kras* [54]. *KRAS^G12D^* induces IL-1α expression via AP-1 activation, leading to NF-κB activation in tumor cells [54]. Activation of NF-κB drives several signaling molecules, cooperates with other signaling pathways including Notch signaling and stimulates PDAC development [55,56]. Activated *Kras* also regulates microRNA’s via NF-κB signaling. Many other oncogenic mutations, such as, EGFR, PI3K and p53, also contribute to NF-κB activation in PDAC [25].

Chronic inflammation can induce NF-κB activation. It has been demonstrated that during chronic pancreatitis, inflammatory cells, and macrophages enhance stroma formation, which increases the risk for PDAC development [40]. Additionally, elevated levels of cytokines and chemokines are observed in PDAC cells, which is correlated with the enhanced NF-κB signaling. For example, elevated levels of chemokine CXCL14, which promotes angiogenesis and tumor growth were found in PDAC [40].

The central region of pancreatic tumors usually contains of a hypoxic microenvironment, which increases the likelihood of angiogenesis, metastasis and drug resistance [57,58]. NF-κB regulates HIF1-α, EMT [59] and angiogenesis factors like VEGF [60] in pancreatic cancer. Fujioka S. et al. demonstrated that inhibiting constitutive NF-κB activity by expressing IκBαM (inhibitor of NF-κB phosphorylation mutant) suppressed liver metastasis in metastatic human PDAC orthotropic nude mouse model. Further, decreases in NF-κB activation reduced expression of a major proangiogenic molecule, vascular endothelial growth factor (VEGF) which shows that inhibition of NF-κB signaling can suppress the angiogenesis and metastasis of pancreatic cancer [60].

## 5. NF-κB in Drug Resistance

The biggest problem in the treatment of advanced pancreatic cancer is the acquired resistance against Gemcitabine [61] and NF-κB was shown to induce chemoresistance to gemcitabine in multiple ways. Studies have demonstrated that human equilibrative nucleoside transporter 1 (hENT1) and human concentrative nucleoside transporter 1 and 3 (hCNT1 and hCNT3) metabolize Gemcitabine. Furthermore, the expression of hENT1 and hCNT3 is associated with chemo sensitivity and overall worse survival in pancreatic tumors [62]. Interestingly, MUCIN 4 negatively regulates expression of hCNT1 transporter through NF-κB signaling pathways, which leads to gemcitabine resistance [63]. Also, cross talk of NF-κB with other pathways leads to gemcitabine resistance. Recently, the role of miR-1266 is explored in the chemo resistance of gemcitabine. How miR-1266 regulates chemoreistance is not clear, however it has been reported that miR-1266 negatively regulates the inhibitors of STAT3 and NF-κB pathways such as SOCS3, PTPN11, ITCH, and TNIP1, which results in constitutive activation of NF-κB and STAT3. Additionally, overexpression of miR-1255 has been linked with chemo resistance of gemcitabine in pancreatic cancer cells [64]. Downregulation of NF-κB/STAT3 signaling axis by Nexrutine1 (reduces transcriptional activity of COX-2) increased gemcitabine chemosensitivity in PDAC cells. In addition to crosstalk between pathways, NF-κB’s role in regulation of stemness also contributes to gemcitabine resistance [65]. Zhang Z et al. reported that pancreatic cancer stem cells develop chemoresistance against gemcitabine by activating NF-κB/STAT3/Nox/ROS signaling pathway [61]. These studies clearly suggest that the blocking of these signaling pathways along with gemcitabine may be a better treatment regimen [61]. Additionally, inhibition of p65, reduces NF-κB activity and consequently inhibits Bcl-2, cyclin D1 and VEGF, and activation of caspase-3. Gemcitabine works synergistically with p65 siRNA and inhibits the proliferation of pancreatic cancer cells. It also suppresses the growth and angiogenesis of pancreatic tumors in animal models [66]. Furthermore, NF-κB induces gemcitabine resistance by interacting with novel regulators, Tripartite motif containing 31 (TRIM31) which is a newly identified E3 ubiquitin-protein ligase, markedly upregulated in pancreatic cancer cell lines and tissue, that correlates to aggressive behavior and poor prognosis in pancreatic cancer patients. Interestingly, the overexpression of TRIM31 mediates gemcitabine resistance via NF-κB activation, and similarly the inhibition of TRIM31 potentiates the cytotoxic potential gemcitabine both in vitro and in vivo model of pancreatic cancer cells in NF-κB dependent manner [67]. Additionally, combined activation of NF-κB and HIF1-α, were shown to induce hypoxia-induced EMT and resistance to gemcitabine [59].

Aberrant NF-κB signaling is also involved in chemoresistance to platinum agents and topoisomerase inhibitors, part of FOLFIRINOX regime, which are standard of care for PDAC. AsPC1-R and BxPC3-R, resistant cell lines against cisplatin, were developed from the PDAC parental cell lines AsPC114 and BxPC3. Pathway Enrichment Analysis (PEA) for genes upregulated in AsPC1-R cells showed dysregulation of NF-κB signal transduction pathway [68]. Furthermore, TGF-β-activated kinase-1 (TAK1) which activates NF-κB and AP-1, has been linked to chemoresistance, because the knockdown of TAK1 in PDAC cells significantly lowered NF-κB activity. Moreover, the addition of TAK1 inhibitor LYTAK1 potentiates the cytotoxic response of oxaliplatin, SN-38, and gemcitabine in PDAC cells [69].

Taken together, the main culprit in chemo resistance is the constitutive activation of NF-κB. Thus, simultaneous targeting of NF-κB signaling pathways along with the chemotherapeutic agents may be a better option in treatment of PDAC.

## 6. Cross Talk of NF-κB with Other Signaling in Pancreatic Cancer

The samples of most pancreatic cancer patients (67–70%) show constitutive activation of NF-κB [40,70]. It is also reported that NF-κB signaling pathways interact with several other signaling pathways (Figure 2). For example, it has been shown that *KrasG12D* mutation, which is seen in almost all patients, is the main driver of constitutive NF-κB expression. How *Kras* mutation drives the constitutive expression of NF-κB has been explored, and it has been shown that cytokine interleukin-1α (IL-1α), mediates this process by activating transcription factor AP-1 [71]. Furthermore, recent studies suggest that cross talk of Notch and NF-κB signaling regulates inflammatory processes in transformed cells [56].

NF-κB and Notch signaling pathways are activated in many cancers, including pancreatic cancer [72,73,74,75]. Activation of Notch signaling leads to proteolytic cleavage of Notch and the release of Notch intracellular domain (NICD), which can translocate and interacts to Rbp-j a DNA binding protein. The nuclear interaction of NICD to Rbp-j regulates gene expression of transcriptional repressors Hes and Hey [76], which are upregulated in PanINs and PDAC. Furthermore, Hes1 regulates expression of nuclear receptor PPARγ. Studies of Maniati et al., suggested that in pre-malignant pancreatic epithelial cells PPARγ is suppressed by the complex of Tnf-α/Hes1. Thus, coordinated activities of NF-κB, Notch and PPARγ regulates key events which play critical roles in the etiology of pancreatic cancers. Along these lines, it has been demonstrated that y-secretase inhibitor which is inhibitor of NF-κB and Notch pathways, suppressed inflammatory genes expression in transformed cells [56]. These findings clearly demonstrate that cross talk of NF-κB, Kras, and Notch signaling pathways governs the chronic inflammatory pathway and eventually promote tumorigenesis [40].

The activation of transforming growth factor- β (TGF-β) exhibits its growth inhibitory effects in normal epithelial cells, including pancreatic cells, yet it exhibits its growth promoting effect in many neoplastic transformed cells, including pancreatic cancer cells. Importantly, it has been established that TGF-β activates NF-κB and plays a key role in the development of PDAC. For example, NF-κB activated by TGF-β suppresses expression of *PTEN* in pancreatic cancer cells [72]. Suppression of *PTEN* by TGF-β could enhance pancreatic cancer molality and facilitate the metastasis. Therefore, activation of NF-κB mediate both TGF-β induced cell motility and suppression of *PTEN* expression. In conclusion, blocking NF-κB by small molecules would elevate the PTEN expression levels and reduce the pancreatic cancer burden.

Varieties of anti-inflammatory and anti-carcinogenic phytochemicals activate Nrf-2 by inducing oxidative stress and lead to the inhibition of NF-κB signaling [73,75]. Curcumin, a PDAC chemosensitizer, regulates NF-κB and Nrf-2. Recent evidences suggested that epigallocatechine-3-gallate exhibits its anti-inflammatory activity through NrF-2 mediated NF-κB inhibition [71]. Further, several Nrf-2 activators include Phenethyl isocyanate (PEITC), SFN and curcumin attenuate liposaccharide induced NF-κB activation [74]. Furthermore, PEITC, and SFN treatment inhibit the phosphorylation of IKK/IκB, nuclear translocation of p65 NF-κB, and inhibit NF-κB signaling. On the other hand, it was reported that NF-κB directly repress the Nrf-2 signaling at the transcriptional level, suggesting the potential cross talk between Nrf-2 and NF-κB [71]. However, it is also unclear whether the failure of clinical trials in the PDACs is because of the Nrf-2 activation by many of these molecules. Therefore, it would be interesting to further evaluate whether using an Nrf-2 inhibitor can improve the outcome in NF-κB inhibiting strategy.

## 7. NF-κB, An Excellent Target for Pancreatic Cancer Prevention and Therapy

A mammoth amount of information suggests that both classical and non-classical NF-κB signaling plays an important role in pancreatic cancer [40]. Studies have demonstrated that the constitutive activation of NF-κB leads to the promotion of cell proliferation, angiogenesis, invasion and metastasis [41,60,76]. Thus, targeting of NF-κB is important in order to prevent pancreatic cancer [40]. Along these lines, several attempts have been made to inhibit NF-κB. Below are the main agents that have been used to inhibit pancreatic cancer.

### 7.1. Cyclooxygenase Inhibitors

A variety of COX inhibitors have been used to suppress pancreatic cancer growth [77]. One of the popular COX inhibitors, aspirin, blocks adenosine triphosphate binding to IKKβ, thus, NF-κB activation [78]. Celecoxib also has some effect on NF-κB, however it has a differential response on pancreatic cancer cells. For example, when celecoxib was used in combination with gemcitabine it reduced the activation of NF-κB in BxPC-3 cells but had no effect on Paca-2 cells [79]. Another COX inhibitor, sulindac, inhibited NF-κB and pancreatic cell growth when used in combination with parthenolide [80]. Sclabas et al. demonstrated that aspirin could be used as a chemo-preventive agent [81]. These studies suggest that NF-κB is a potential target for cancer prevention and COX inhibitors could be utilized as chemo-preventive agent.

### 7.2. Curcumin

Curcumin is a natural yellow compound found in turmeric. Almost 4000 articles have been published on Curcumin, and some of the studies have named this molecule as a “magic bullet” [82,83,84]. Curcumin exerts its anti-cancer activities through a variety of mechanisms including inhibition of NF-κB [85,86]. Several studies have demonstrated that Curcumin shows its anti-cancer and anti-inflammatory activities by inhibiting both IKK and NF-κB activity in a variety of cancer cells, including pancreatic cancer [74,87,88]. A significant amount of studies has demonstrated that Curcumin inhibits NF-κB and promotes apoptosis in a variety of pancreatic cancer cells and orthotropic model [74,88]. Recently, a phase 2 clinical trial was completed at MD Anderson Cancer Center, and the study concluded that oral curcumin is well tolerated and exerts biological activities in pancreatic cancer patients [89].

### 7.3. Other Agents

Several other NF-κB inhibitors have shown great potential to inhibit pancreatic cancer but have limited studies. Some of these agents are protein kinase C inhibitors, parthenolide, polyphenols, protein tyrosine kinase inhibitor, GSK-3 kinase inhibitor, proteasome inhibitors, and IκBα mutants [77]. As drug discovery and cancer research progresses, we anticipate the discovery of more exciting inhibitors of NF-κB in order to combat pancreatic cancer.

## 8. Conclusions

Several lines of evidence suggested that both canonical and non-canonical NF-κB activation is responsible for PDAC progression and metastasis. Further, the cross-talk of NF-κB with other signaling pathways indicates that it is the cellular wiring which is critical in development of PDAC. Targeting of this signaling pathway is important in order to prevent PDAC. As described above, there are a variety of NF-κB inhibitors which show great promise and there are many more to be discovered. However, despite the effectiveness of these molecules in laboratory studies, only few molecules are in clinical trials, thus more studies on patient samples are needed. In addition, several studies have demonstrated that combination therapy could be beneficial in order to treat pancreatic cancer or prevent the chemotherapy resistance to pancreatic cancer.

Overall, the mammoth studies of NF-κB and its relationship with pancreatic cancer have laid a solid foundation for drug development. It is an opportune time for the development of novel agents that target these signaling pathways. Several agents have been discovered already and many more are in the pipeline. However, extensive studies on animal models followed by clinical trials are needed to fully understand the mechanism of these drugs.

## Figures and Tables

**Figure 1 ijms-19-03890-f001:**
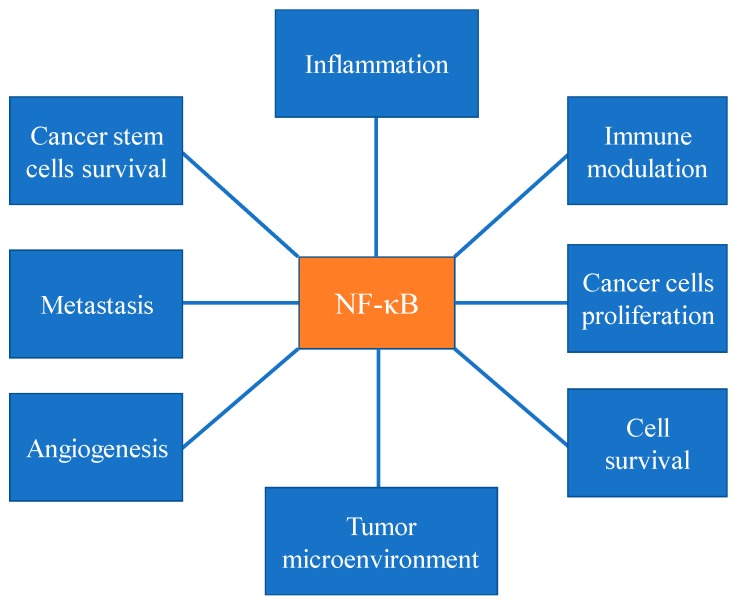
Different cellular events, including inflammatory response, immune cells modulation, cancer cell proliferation, cancer cell survival, tumor microenvironment, angiogenesis, metastasis, stem cell survival activates NF-κB pathway.

**Figure 2 ijms-19-03890-f002:**
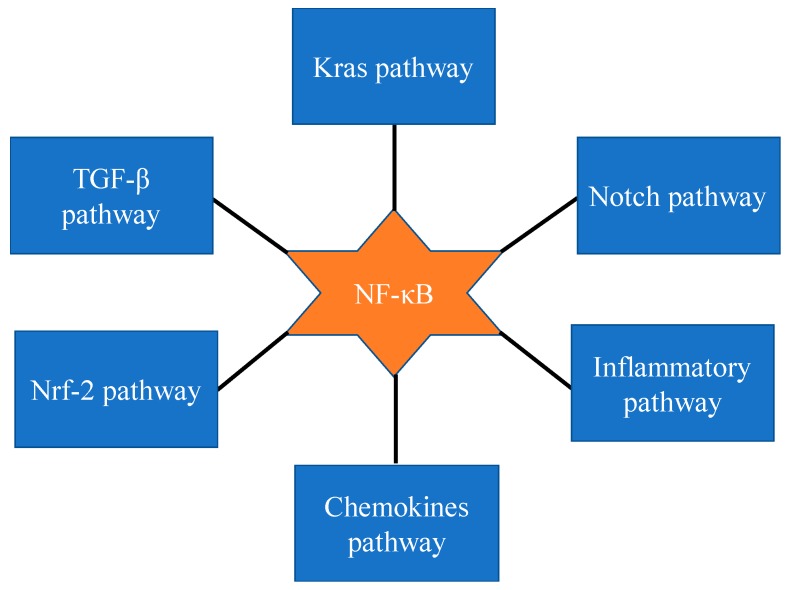
Cross talk of NF-κB signaling with others signaling in pancreatic cancer.

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
