# Peer review of "Advancement of NF-κB Signaling Pathway: A Novel Target in Pancreatic Cancer"

_ijms, 2018, doi:10.3390/ijms19123890_

Round 1

Reviewer 1 Report

In the current review manuscript, the authors Pramanik et al. try to discuss the NF-kB signaling pathway in pancreatic cancer and therapy. The article sounds interesting but there are still some concerns to be addressed:

Major concerns:

1. Please describe p105 and p100 processing briefly.

2. Some information about the IKK complex need to be added. The authors are mentioning NIK as a key kinase in the non-canonical NF-kB pathway in 1.3 and IKKb in 4.3. Brief information about the IKK complex (and NIK) is helpful for readers.

3. There are many excellent publications with pancreatic cancer and NF-kB deficiency mouse models (e.g. IKKg/NEMO knock-out with KRASG12D) which addressed also molecular characterization. The authors are asked to cite and discuss some of the papers.

4. The authors mention in the chapter 3, “the alternative pathway… regulates… tumor microenvironment”. In the chapter 4.4 “tumor microenvironment”, the alternative pathway is not discussed. The authors are requested to briefly mention what the alternative/non-canonical NF-kB pathway does for tumor microenvironment.

5. The title is “a novel target in pancreatic cancer”. However, the authors in the chapter 8 mentioned only COX inhibitors, curcumin etc. Are they all? Since pancreatic cancer prevention and therapy are expected to be a major issue of the paper, the authors are required to improve the chapter.

Author Response

We would like to thank reviewers and the editors for a candid review of our manuscript entitled “Advancement of NF-κB Signaling Pathway: A Novel Target in Pancreatic Cancer”, which now has been extensively revised in accordance with the comments and suggestions of the reviewers. The revised manuscript is significantly better from the original version due to inclusion of new information, majority of which was generated in direct response to the reviewer’s comments. Clearly, the changes made in response to the reviewer’s comments have improved the quality of this revised manuscript. Our point-by-point response to the reviewer’s specific comments, and the changes made during revision are detailed below and the changes in the manuscripts are highlighted in yellow with track change. Due to the technical problem in endnotes, we have also attached the accepted track change manuscript with the updated references. We hope that the revised manuscripts would now be accepted for publication in IJMS.

Thanking you.

Sincerely,

Kartick

Response to Reviewer # 1:

1. We have now added briefly about the processing of p105 and p100 with proper references.

2. Authors are thankful for this suggestion. We have now added briefly about IKK and NIK complex.

3. We have now cited some papers and discussed in appropriate section.

4. We have now added the information about the non-canonical NF-κB activation in tumor microenvironment.

5. We agree with reviewer regarding the information on novel agents. We intentionally did mention these agents in the subsection of other agents. We only mentioned about the agents those are well established. We carefully analyzed other agents and we found that either other novel agents are in very early phase or they have some limitations. Thus we did not mention in this review. Hopefully, when we write next review, few novel agents will be established.

Reviewer 2 Report

The review manuscript highlighted the role of NF-KB signaling in pancreatic cancer with special focus on therapeutic strategies. The review provided interesting insights in the mechanism of action of NF-κB signaling and how it can be targeted in pancreatic cancer. However, substantial changes are recommended to make the manuscript up to date and interesting to the readers.

Comments and suggestions for authors

1. The authors are recommended to add introduction section in the manuscript.

2. Lines 173-174: Figure 1 is not required here.

3. Lines 177-180: “Inhibition of NFκB activity inhibits the tumor growth and angiogenesis, through downregulation of the angiogenic molecules VEGF and IL8 in glioblastoma cells”. Is NF-κB effect in angiogenesis seen only in glioblastoma cells.

4. Lines 184-186, “Blocking NF-κB inhibited the transcription of the EMT genes CDH2, SLUG, TWIST1 and SNAIL in a metastatic breast cancer cell line”. Is breast cancer cell line the only metastatic cancer cell line showing this effect.

5 Lines 226-229: Reference needs to be added here.

6. Lines 228-229: The authors are recommended to mention the status of the overall survival, either worse or improve.

7. Line 267: Add approximate percentage of pancreatic cancer patients that show constitutive activation of NF-κB.

8. Line 302: Add the full form of SFN.

9. Lines 304-306: Reference needs to be added here.

10. Line 307: “It also remains vague that failure of clinical trials in PADACs is because of Nrf2 activation by many of these compounds”. The sentence needs to be reframed

11. Line 322: Is Paca2 cells same with MiaPaCa2 cell line? Recommended to use the full name.

12. The authors are recommended to confirm that only abbreviations used in the manuscript are included in the abbreviation section.

13. The authors are recommended to add more references.

14. The authors are recommended to check for grammatical errors and to look for the flow of the manuscript.

Author Response

Response to Reviewer # 2:

1. We would like to thank the reviewer for the critical evaluation of our review manuscript and providing constructive suggestions. As suggested by the reviewer, we sum-up the number 1 and 2 headings and replaced with introduction. 

2. As suggested by the reviewer, we removed the “figure 1” word from line 173-174.

3. We totally agree with the reviewer comment. As suggested by reviewer, we have incorporated effect of NF-kB in angiogenesis with other cancer study in the revised manuscript.

4. We also agree with the reviewer comment. As suggested by reviewer, we have incorporated the other study in different cancer models with effect of “Blocking NF-κB inhibited the transcription of the EMT genes.

5. As suggested by the reviewer, we incorporated the reference.

6. We would like to thank the reviewer for the critical evaluation of our manuscript and providing constructive suggestions. Higher expression of hENT1 could worsen the overall survival of tumor cells. As suggested by the reviewer, we incorporated status “Worse” of the overall survival.

7. About 67-70% pancreatic cancer patients show constitutive activation of NFKB. We incorporated this information in the revised manuscript.

8. As suggested by the reviewer, we incorporated the full name of SFN in abbreviation section. SFN – Sulforaphane

9. As suggested by the reviewer, we incorporated the references for 304-306 line in the revised manuscript.

10. As suggested by the reviewer, we rephrased the sentences as “However, It also unclear that failure of clinical trials in the PDACs is because of Nrf-2 activation by many of these molecules”

11. Yes. Paca-2 cells same with MiaPaca2. They are pancreatic cancer cell lines.

12. As suggested by the reviewer, we have included all abbreviations in the abbreviation section.

13. As suggested by the reviewer, we have incorporated more references in the manuscript.

14. Thank you very much for this constructive suggestion. As suggested by the reviewer, grammatical errors in the manuscript have been checked by scientific editorial department.

Round 2

Reviewer 1 Report

The authors Pramanik et al. took into account the comments from the review for the manuscript entitled "Advancement of NF-κB Signaling Pathway: A Novel Target in Pancreatic Cancer" and they proceeded with the necessary revisions.

Reviewer 2 Report

In the updated version of the review manuscript titled "Advancement of NF-κB Signaling Pathway: A Novel Target in Pancreatic Cancer" the authors have taken into accounts the reviewers' comments and incorporated the recommended corrections. The review would be interesting to readers in cancer biology.

Best of luck.